# Navigating Voice, Vocabulary and Silence: Developing Critical Consciousness in a Photovoice Project with (Un)Paid Care Workers in Long-Term Care

**DOI:** 10.3390/ijerph19095570

**Published:** 2022-05-04

**Authors:** Saskia Elise Duijs, Tineke Abma, Janine Schrijver, Zohra Bourik, Yvonne Abena-Jaspers, Usha Jhingoeri, Olivia Plak, Naziha Senoussi, Petra Verdonk

**Affiliations:** 1Amsterdam UMC, Department of Ethics, Law and Humanities, Amsterdam Public Health Research Institute, De Boelenlaan 1089a, 1081 HV Amsterdam, The Netherlands; zobourik@gmail.com (Z.B.); yvonneabena@icloud.com (Y.A.-J.); ujhingoeri@hotmail.com (U.J.); f.plak@planet.nl (O.P.); naziha_senoussi@nisa4nisa.nl (N.S.); p.verdonk@amsterdamumc.nl (P.V.); 2Department of Public Health and Primary Care, Leiden University Medical Center, Albinusdreef 2, 2333 ZA Leiden, The Netherlands; abma@leydenacademy.nl; 3Leyden Academy on Vitality and Ageing, Rijnsburgerweg 10, 2333 AA Leiden, The Netherlands; 4Independent Researcher/Photographer (Scholarly Artist), Sichting B.a.d., Talingstraat 5, 3082 MG Rotterdam, The Netherlands; janineschrijver@xs4all.nl

**Keywords:** participatory health research, photovoice, critical consciousness, community participation, intersectionality, epistemic justice, long-term care, unpaid care workers, paid care workers, occupational health

## Abstract

Photovoice is a widely used approach for community participation in health promotion and health promotion research. However, its popularity has a flip-side. Scholars raise concerns that photovoice drifts away from its emancipatory roots, neglecting photovoice’s aim to develop critical consciousness together with communities. Our four-year photovoice project aimed to unravel how the health of (un)paid care workers was shaped at the intersection of gender, class and race. This article springs from first, second and third-person inquiry within our research team of (un)paid care workers, academic researchers and a photographer. We observed that critical consciousness emerged from an iterative process between silence, voice and vocabulary. We learned that photovoice scholars need to be sensitive to silence in photovoice projects, as silence can be the starting point for finding voice, but also a result of silencing acts. Social movements and critical theories, such as intersectionality, provide a vocabulary for participants to voice their critical perspectives to change agents and to support collective action. We discuss our experiences using Frickers’ concept of ‘epistemic justice’, arguing that critical consciousness not only requires that communities are acknowledged as reliable knowers, but that they need access to interpretative tropes to voice their personal experiences as structural.

## 1. Introduction

Photovoice is a widely used approach for community participation in health promotion and health promotion research [1]. Photovoice is a visual methodology that aims to foster collective action and social change [2,3]. It is often conducted within participative and action-oriented research approaches, such as Participatory Health Research [4], Participatory Action Research [5], Community Based Participatory Research [6] and Transformative Research [7]. 

The goal of photovoice is threefold [2]. First, photovoice allows participants (or: community members) to express their lived experiences by using photography. Second, photovoice aims to create a dialogical space to understand how personal experiences are shaped by broader structural inequalities. Third, it enables participants to challenge these experiences of marginalization together with researchers and other stakeholders (“change agents”) such as professionals and policy makers. 

Photovoice, as developed by Wang and Burris, is grounded in critical social theory, including feminism, postcolonialism, social justice theory and Paolo Freire’s work on the pedagogy of the oppressed [2,3,8]. In the last decades, photovoice became increasingly popular among health scholars [9]. Currently, photovoice is often seen as a “fun”, “quick” and “easy” way to gain access to participants lived experiences and share these in an accessible manner with policy makers [3]. However, its popularity also has a flip-side. Scholars raise concerns that photovoice drifts away from its critical and emancipatory roots, neglecting photovoice’s aim to critically reflect upon broader societal structures of oppression together with participants [10,11,12,13]. 

Liebenberg [3] (p. 1) brings into remembrance how Wang and Burris aimed photovoice to be an *“analytical, pro-active and empowering”* endeavor that honored participants’ expertise and wisdom. She argues that “*part of honoring this wisdom and experience requires us to facilitate critical reflection on structurally embedded experiences, and that the knowledge emerging from this reflection is both given a platform from which to be voiced, and equally important, amplified in ways that are heard*”. In other words, photovoice is not solely meant to voice participants’ lived experiences to policy makers. Photovoice should also amplify participants’ perspectives on structural systems of oppression to these policy makers. Failing to address these structural issues could—unintentionally—turn photovoice projects into a disempowering experience [14].

However, knowledge about developing critical perspectives on structural systems of oppression is underrepresented in photovoice literature [3,11,12,13]. Multiple (scoping) reviews—in diverse fields—focus on scholars’ difficulties to reach policy makers with outcomes of their photovoice project, but they pay little attention to the critical reflection of participants [15,16,17,18,19,20,21,22]. Articles that focus on developing critical consciousness with participants remain scarce [23]. 

In this article, we describe how critical perspectives on structural embedded experiences can be developed in dialogue with participants rather than about participants. We draw from our experiences in a four-year photovoice process with paid and unpaid caregivers in residential long-term care. These paid and unpaid caregivers are members of the health workforce in long-term care and, thus, a part of the community (they are not representatives of this community, as no citizen can ever represent its’ entire community) of (un)paid care givers. In this project we critically reflected upon community researchers’ experiences with care work, informal caregiving and health and collectively unraveled how these were shaped by structural inequalities, including gender, class, ethnicity and age. This photovoice project was part of a bigger PHR study into the health and wellbeing of (un)paid care workers. Our photovoice project was informed by critical gender and intersectionality theory. We will elaborate on intersectionality in more detail below (theoretical framework). We reflected upon our four-year project with the entire research team, consisting of academic researchers, a scholarly artist (photographer) and five (unpaid) care workers who participated in the photovoice process, to whom we will refer to as community researchers. 

In our photovoice project, we collectively unraveled the complex interplay between gendered (theme 1), classed (theme 2) and racialized (theme 3) inequities. Reflecting upon this process, we came to understand that the critical consciousness of these structural inequities emerged from an iterative process between silence, voice and vocabulary. Based on our reflections, we argue that photovoice scholars need sensitivity to recognize many meanings silence can have in photovoice projects, as these as the starting point for findings voice. We also stress the importance of engaging with critical theories, such as intersectionality, as these provide a vocabulary that enables participants to voice their critical perspectives to themselves and to others. 

We conclude that critical consciousness on structural inequities is an essential aspect of epistemic justice [24]. Facilitators of photovoice can contribute to epistemic justice by creating spaces in which participants are acknowledged as reliable knowers (witnessing justice). However, Fricker’s notion of epistemic justice urges facilitators to introduce interpretative tropes that are necessary for critical consciousness (hermeneutic justice) [24]. 

### Main Objectives

With this article, we aim to address current concerns about photovoice by illustrating how critical consciousness can be fostered in photovoice processes together with participants. In addition, we aim to contribute to the growing body of literature that aims to combine emancipatory research approaches, such as PHR and photovoice with intersectionality [25,26,27,28,29,30] as we think that intersectionality offers a valuable framework for critical reflection on the specific nature of these societal structures of oppression.

## 2. Theoretical Framework

### Intersectionality

In the last decades, intersectionality has emerged as such an ‘interpretive trope’ to understand lived experiences in their socio-political context. Grounded in queer women of color’s experiences and academic reflections, intersectionality has emerged as an ideological, theoretical and methodological approach [30,31,32,33]. It aims to understand how multiple aspects of identity and/or multiple systems of oppression interact with each other to shape peoples lived experiences [34]. The term intersectionality was coined by legal scholar and critical race theorist Kimberlé Crenshaw. Embraced by activist movements, intersectionality builds upon the premise that “all oppression is connected” which provides a shared narrative that allows for alliances between social movements. 

In the last decade, intersectionality receives growing attention from scholars studying (health) inequities [35,36,37]. For health scholars, intersectionality provides a framework to (1) understand health inequities within and between groups [38], (2) identify groups who are specifically at risk, but remain invisible in single-axe analysis [39], and (3) understand how these health inequities are shaped in their broader societal context, taking into account societal systems of oppression, including but not limited to patriarchy, class, racism, ableism or environmental injustice [36].

## 3. Setting

### Societal Context of Our Photovoice Project “Negotiating Health” 

In the Netherlands, the health of paid and unpaid caregivers is under pressure. Due to an ageing population and austerity measures in long term care (LTC) both paid and unpaid caregivers have to deal with higher care loads. Studies show how their health and wellbeing is increasingly under pressure, but not for all care workers in a similar manner [40].

In response, scholars and societal organizations expressed concerns about growing societal inequalities due to these policy transitions. Care organizations wonder how they can provide diversity-sensitive HRM policies to their employees, to address their specific care needs. They advocated for diversity responsive and intersectional perspectives on the health of healthcare workers. 

Therefore, authors of this paper worked together in a (PHR) study called Negotiating Health (2018–2022). This study aimed to understand how the health and wellbeing of (un)paid care workers in elderly care (45–67 years of age) was shaped at the nexus of gender, class, race and disability, thus: from an intersectional perspective. This study was funded by the Netherlands Organization of Health Research and Development (grant number 849200012). The research team consisted of academic researchers (Duijs, 36 years old; Abma, 58 years old; Bourik, 50 years old; Verdonk, 57 years old), a professional photographer (Schrijver, 50 years old) and five community researchers who have experiential knowledge as an (un)paid caregiver in residential long-term care (Senoussi, 55 years old; Abena-Jaspers, 55 years old; Plak, 63 years old; Jhingoeri, 54 years old; anonymous, 55 years old). 

The findings in this paper derive from the (photovoice) process that took place in advance (phase 1), parallel to (phase 3) and after (phase 4) several interview studies that were conducted as part of Negotiating Health (phase 2). The findings from these qualitative sub-studies are published elsewhere [40,41,42,43,44,45]. 

## 4. Methodology

### 4.1. Research Approach

We position ourselves in the critical and feminist strands of PHR [6,25,26,46]. We conducted our photovoice project, following Wang and Burris’ [2] approach, who employed photovoice as a means to foster critical reflection, collective action and social change [3]. We opted for photovoice from our own positive experience with arts-based research [47]. The long-term care sector is characterized by dominant narratives and recurring societal debates around the health and wellbeing of care workers, often focusing on psychosocial working conditions. We aspired to move beyond such well-known narratives and capture new realities, literally, that would spark collective action. Photography has the potential for emotive and moral appeals on change agents [2,47]. 

### 4.2. Phases of Our Research Process

The photovoice process consisted of (roughly) six-weekly, 2 h long meetings between 2018 and 2021. Roughly, four phases can be identified (see Table 1). In the first phase, ten paid and unpaid caregivers in long-term care participated in the photovoice process to set the research agenda for the broader PHR study. In the second phase, five out of ten participants continued to participate as community researchers. Together, we conducted three qualitative interview studies into the experiences of men and women, hired employees and self-employed care workers in long-term care (phase 2). Parallel to these qualitative studies, we continued our own reflexive process through photovoice and aimed to deepen our understanding of the societal and intersecting systems that shaped our own and others’ lived experiences in long-term care (phase 3). In the last phase (phase 4), we co-created four portraits, included in a book, to communicate our main findings to change-agents. 

Our six weekly photovoice meetings were facilitated by the academic researchers and the professional photographer. Data in this photovoice process consisted of the photographs, made by and of participants in co-creation with the professional photographer. Other sources of data are audio-tapes and transcripts of monthly meetings and field-notes of the academic researchers/photographer. In addition, in crucial moments semi-structured interviews and numerous informal conversations took place between researchers, photographer and community researchers. 

### 4.3. First, Second and Third-Person Reflection

PHR scholars stress the importance of first-, second- and third-person inquiry [48]. First person research centers around the reflections of one’s own practices, dilemmas and emotions (I-perspective). Second person inquiry focusses on reflection between people, such as the reflection that takes places within a PHR research team (we-perspective). Third person reflection refers to the reflection that takes place within broader communities, such as a community of practice or academic debates. This article springs from the first, second and third-person inquiry that took place within our PHR research team. We extensively reflected upon our photovoice project during and after the photovoice process, continuously moving back and forth from individual perspectives (first-person), collaborative reflections (second-person) and connecting these to academic theories such as Fricker’s [24] epistemic justice or intersectionality [31,32,33,34,35,36,37,38,39] (third-person). The academic researchers took the lead in writing this article, yet the voices of the photographer and community researchers have a prominent place. The consolidated criteria for reporting qualitative research (COREQ) guided the reporting of this study [49]. In addition, the article was re-written until everybody felt that conflicting and shared perspectives were adequately described. This study was evaluated by the VUMC Medical Ethical Review Committee which confirmed that the Dutch Medical Research Involving Human Subject Act did not apply (dd. 17 April 2018). All community researchers consented to including their pictures in this article. 

## 5. Results: Reflections on Voice, Vocabulary and Silence in Photovoice 

In this section, we present the dynamic between silence, voice and vocabulary in four themes, which reflect the chronological process of our research project:In the first theme, “What on earth am I doing?”, we described how we developed critical consciousness on gendered inequities. This process took place in phase 1 of our photovoice project.In the second theme “We should all be wearing yellow vests”, we describe how critical consciousness led to us speaking out about inequities that were shaped at the intersection of gender and class. This process took place at the beginning of phase 2 of our photovoice project.In the third theme, “You’d rather not see it”, we not only describe how we broke the silence on racial inequities but also how we were consequently silenced by each other and by change agents. This process took place at the end of phase 2 of our photovoice project.In the fourth theme, “What you don’t see”, we present the portraits and booklet that we created and used in dialogue with change agents. These portraits and booklet describe how the health and wellbeing of paid care workers in long-term care is shaped at the intersection of gender, class and race. This process takes place in phase 4 of our photovoice project.

Each theme will reflect how critical consciousness came about in this iterative process of silence, voice and vocabulary. 

### 5.1. ‘What on Earth Am I Doing?’ Developing Critical Consciousness on Gendered Inequities

At the beginning, our photovoice project was experienced as a *preluding silence*: “*a process that allows one to go within before one has to speak or act*” [50] (p. 2). This silence was enabled by photography.


*‘If you make a photo, you literally have to stand still and look at your live. And then it also becomes visible for others. When I looked at my own pictures, I realized: what on earth am I doing? That made me think about my life’.*
(community researcher)

We immediately observed a gendered difference in the photographs. Whereas the women’s photographs were personal and intimate, the male community researcher voiced political concerns about working in long-term care, being initially silent about the impact on his own life and work. The photographs sparked a dialogue among the participants in which they started questioning experiences that were normalized.


*‘You start talking and recognize things from each other. That’s when you start thinking: “this is not normal”.*
(community researcher)

Participants could draw upon unconscious, embodied and affective methods of knowing in the silence that photography enabled. The professional photographer described this process as follows.


*‘Directing the camera towards your own life starts a reflective process. It urges you to think about what you see in the picture. To attribute meaning to it. You often start by what is unconsciously captured in the picture, by what is ‘inside’ you. It’s like soil. You start digging, shuffle and wield the ground, and allow for the air to come in’.*
(photographer)

Participants started voicing their experiences to each other (Figure 1a–d). They felt literally seen and acknowledged by each other. This experience contrasted with their experiences as care workers, which they characterized as, “*Being invisible, doing hidden work*”. They voiced this insight through a poster we co-created at the end of phase 1.

For academic researchers, participants’ photographs and lived experiences with “*being invisible, doing hidden work*” resonated with academic theories on women’s’ caring identities [51,52] and women’s care responsibilities as a source of gender inequity [53,54,55,56,57]. The academic researchers shared these reflections with the participants but noticed that these were not (yet) engaging for participants. Feeling pressured to publish about our project, the first author published an article in which she theorized about the participants, rather than with the participants [40]. Opening up a conversation about which theories did resonate, and which did not, enabled us to search for a vocabulary that enabled participants to understand their experiences. Participants emphasized the meaningfulness of caregiving [Figure 2] and the pain of not being able to care (Figure 3). 

Their experiences did resonate with a care ethical perspective [56,57] that emphasized the importance of caregiving for our societies. Looking back, community researchers appreciated that theoretical concepts were brought into the conversation, even when they did not all resonate with (all of) them: Theories provided had the potential to break the silence also in one’s self.


*‘For me, these moments were really eye-opening. You just realize, wow, what happened to me (pregnancy discrimination) was not just an incident. I started to look back at certain life events, and started to see them in a new light.’*
(community researcher)


*‘If you don’t have the words to describe what happened to you, then how can you speak about it? If it’s something that is never spoken about, you don’t hear the experiences of others, and you are the first to put it into words, that is just so hard. It’s not likely that it will surface, or that you will speak out.’*
(academic researcher)

Engaging with theories (vocabularies) did result in tensions and dilemmas also within the research team. The academic researchers and the photographer struggled with the following question: “When and what is an ethically and relationally sound way to engage with complex concepts in a photovoice process?” While some believed that engaging with theories was essential for fostering critical consciousness, others feared that adding theoretical perspectives by academic scholars might endanger participants own process of finding voice. We concluded that engaging with theories requires courage, sensitivity and timing from facilitators to bring this this knowledge to the table in a horizontal manner.


*‘It’s a process of trial and error, and that’s no problem as long as you stay connected with each other throughout this tension. If participants don’t respond to our input, we can reflect on it, maybe they need time to process, maybe they cannot relate. Either way, it requires courage to stay close to ‘what is’, that is the hardest thing to do.’*
(photographer)

### 5.2. “We Should Be Wearing Yellow Vests!” Speaking out about Inequity at the Intersection of Gender and Class 

In the second phase, community researchers’ (from here onwards, we speak of community researchers rather than of participants to emphasize the changing nature of our collaboration. See Table 1) experiences as low-paid workers in residential long-term care were discussed more elaborately. The community researchers began to understand some of their experiences with (paid) care work as a form of gendered and classed-based exploitation. Obviously, it was not the gendered caring responsibilities per se but rather the fact that care work was structurally devalued, also economically, by society, political policies and healthcare institutions, putting their health, wellbeing and financial situation under pressure [53,54,55]. This understanding sparked anger in the group and immediately evolved into collective action. The community researchers found their voice. What happened?

At this time, the French “Yellow Vests movement” was prominently covered in newspapers and television. People protested against low wages in combination with increasingly high costs of living. In a meeting, we discussed the impact of paid and unpaid care work on community researchers’ health and livelihood. This issue was of particular importance, as we observed that many of care workers deal with poverty and/or debt in our interview studies. One of the community researchers repeatedly started saying, “*You know, we should also be wearing Yellow Vests*.” The others agreed—the vocabulary of class-based inequities resonated with them. Two community researchers had formerly been active in the labor union and/or in local politics, and they took “ownership”, as they thoroughly understood what was at stake and embraced class in particular as an “interpretive trope” (vocabulary).

An academic researcher (first author) responded by asking what the Yellow Vests should look like and what slogans would fit. Slogans emerged at the interaction of gender and classed-based exploitation, “*I provide care with love, but not at the expense of my income, pension and health*”. The vests were printed, and in the next meeting, the community researchers’ picture was made as an official “protest” (Figure 4 and Figure 5).

As critical consciousness on gender/class-based inequities grew, the community researchers wanted to voice their message to a broader public. Collaboratively, we translated our insights and righteous anger into an op-ed to be published in a national newspaper. In the op-ed, the community researchers stated that quality of care came at the expense of their health, income, free-time and pensions [41].

In addition to the op-ed and the “protest” pictures, the community researchers also wanted to be photographed by the professional photographer with their Yellow Vests and with their protest signs (Figure 6 and Figure 7). Photography became an act of protest in itself, and an act of making the invisible—literally—visible. This illustrated the pivotal role of photography, and other arts-based methods in this project. Photovoice allowed for both “verbal” and “non-verbal” vocabularies and speaking up was supported by a variety of ways to express voice, including art. By “commissioning” these portraits, they took control over how they wanted to be photographed, which reflected their empowerment process. 

Although the op-ed was written collectively, several community researchers did fear for the consequences at their workplace by speaking up (silence). Eventually, the op-ed was signed by the only male participant. Although he was just as anxious as the other community researchers, he felt backed up by the union, his manager and the fact he recently exited LTC to work in psychiatric care. His sent his reflections on his individual decision-making process on whether to co-sign the op-ed to us and to his manager.


*“And suddenly I was very ashamed, that in these peaceful times, I am afraid to put my name under a newspaper op-ed. Notably, the resistance newspaper that my father risked his life for many years ago.”*
(Community researcher)

His letter illustrates how the ability to speak out (voice) was shaped at the intersection of gender and class, in this case professional status. Expressing criticism on the “system” without fearing consequences or backlashes was an issue for everyone, including the academic researchers. By supporting the community researchers’ op-ed, the latter put their position as “neutral” and “objective” researchers on the line. This concern was explicitly voiced by a member of the project’s steering group. In the end, the op-ed was signed by the full professor (author) and by the male community researcher; those with the most privileges. 

Amplifying the community researchers’ critical perspectives to places outside of the direct research process was challenging. The op-ed received support from care workers and unions [58,59], but it was critically appraised by our own steering group. Several members fed back that our op-ed focused “solely on what was negative”, that it was not representative for the entire long term care sector, that it reinforced negative stereotypes about working in long-term care and that it was not solution-oriented and failed to offer practical recommendations or clear-cut policy directions. Their responses communicated to the community researchers that there was limited space for their authentic “rough-edged” narratives, although they also understood that this response came from a place of concern and worry for the sector. Throughout the project, we have not succeeded in organizing a dialogue between the steering group and the community researchers. Fear of being silenced was an obstacle for the community researchers to take a seat at the table. 


*“This project enables to voice my concerns, speak out about the things that need to be heard. Our group makes me feel safe, and this safety helps you to articulate your experiences and voice them to others. I cannot always deal with the confrontation with managers or policy makers. They make you feel so small, so powerless. You don’t expect any support. They will downplay your story: “It’s not representative for the entire sector. It’s not happening. It’s not true.” And then you start believing, maybe you are right.”*
(Community researcher)

### 5.3. “You’d Rather Not See It” Breaking the Silence on Racism

In the third phase, racism and its impact on (occupational) health emerged from interviews with other care workers. These interviewees’ stories uncovered (community) researchers’ own experiences, experiences that had thus far remained unspoken (silence). We spoke of experiences with race in general, but particularly with racism shaping their daily care work and labor market experiences in the care sector. Our photovoice project did not “educate” the community researcher on matters of “race”. Obviously, and unknown by the academic researchers, most community researchers had already been racially aware, and they were urged by their children to speak out.


*“We have learned to stay silent. Be modest. Listen to your boss. My children are not like that. They would say to me: mom, speak out! The younger generation doesn’t stay silent, like we have learned to be.”*
(Community-researcher)

Rather, we reached a level of trust that allowed community researchers (and academic researchers) to express these vocabularies. The community researchers entered our photovoice project with a fair dose of “healthy mistrust”, as to them, it was not to be expected that white academic researchers would acknowledge their knowledge on racism as trustworthy or would support them in speaking out about these issues.


*“Too often you are asked to participate in a research project that turns out to be about someone else’s agenda. They say it’s about us, but in the end, it is not about us at all. (…) Our stories are often just erased.”*
(Community researcher)

Many stated that they were urged by their children to speak out, “*The younger generation doesn’t stay silent, like we have learned to be*” (community researcher). Moreover, in our project, community researchers and academic researchers started to voice their experiences. Speaking out was supported by arts-based methods. Together, we watched a video “Variations on White”(password: White) on racism in healthcare, developed in an earlier project at our department, which supported the conversation. Meanwhile, in the outside world, the Black Lives Matter movement placed racism front and center in the nation’s attention. BLM provided an opportunity for white academic researchers to openly identify themselves as an ally, by supporting the protests and speaking out in public [60]. In our app group, countless articles and experiences were shared about racial inequities, reflecting the group’s critical consciousness of racism. 

However, while class-based inequities united the group and created collective action, the racialized discussion caused friction, especially in relation to the white male community researcher. Racism was a contested vocabulary, and as the women in the group started sharing feminist and anti-racist outings in the group app, the white male community researcher distanced himself from the group. 


*“I decided to quit with our project. I realized that I was very angry about something (…) Now, I realize that I feel attacked as a “white man”. It is very unpleasant to be held accountable as a member of a group, when this group is seen as something that is very different from who I am and how I see myself.”*
 (Community researcher)

Critical consciousness about gender/class moved our focus to broader societal structures, enabling a conversation without blaming specific individuals. However, the opposite happened with racism. The discussion immediately became “personal”. Conversations did not lead to collective action, but they were experienced as divisive by the male community researcher. He felt misunderstood and excluded. We were not able to collectively speak of racism as a system that hurts us all; the system was exclusive and the vocabulary used to expose the system was not. The other community researchers did not respond to him leaving the group. Again, silence entered our joint space.

Silence on racism entered our photovoice process multiple times hereafter. For example, in a session that aimed to visualize the main findings of our study, the photographer—who had not been part of all our conversations about racism and aimed to explore participants’ experiences in the midst of the COVID-19 pandemic—glanced over the main themes, stumbled upon the word “racism” and, in a split second, discarded the theme, “*hmm, that might be a little too abstract for now*”. None of us intervened. Only later, we began to understand this dynamic as result of whiteness. Racism can only be “*too abstract*” from the perspective of those who do not have to live the experience of racism. Fortunately, we had built enough trust to reflect on this moment and, in a humbling process, we broke our white silencing and brought racism back to the table. 

Silence also happened in dialogue meetings with change agents as part of our photovoice process. For example, in a meeting with managers, one manager expressed her love for the photograph of a community researcher that expressed the necessity to root out racism. After the manager listened to the corresponding narrative and became aware of the message, she stated the following.


*“Now I know the story behind the photograph, it is not as pretty to look at anymore. You’d rather not see it”.*


For the community researchers, such reactions reflected their positions within many LTC organizations.


*“Our stories have a rough edge. You can consider it a bad thing, but it is what it is. It’s not like social media, where everything is covered up under a nice filter. This is reality. Our reality. A lot of people live in a different reality. Then our stories might be too rough and confronting. Not everybody is willing to look at it.”*
(Community researcher)

### 5.4. “What You Don’t See” a Book Voicing Critical Perspectives on Gender, Class and Race

In this paper, we describe the critical perspectives on gender, class and race that emerged from the iterative process of silence, voice and vocabulary. This process resulted in four portraits of the community researchers, each communicating a critical perspective on how their health and wellbeing as paid care workers in long-term care is shaped by structural inequities (Figure 8a–d). These portraits are used in dialogue meetings with change agents, such as managers in LTC, policy makers, HRM managers and occupational health professionals and will be published in a book that is co-created by the community researchers, academic researcher and photographer. Our book presents the iterative process between silence, voice and vocabulary in an artful and conceptual manner. This book invites readers to break the silence as they have to actively uncover the four portraits (Figure 9). The portraits capture the broader structural issues, and the reader can listen to the narratives of the community researchers speaking out about these injustices (Figure 9).

In doing so, we aimed to fulfill photovoice’ ambition, which requires “that the knowledge emerging from this reflection [on structural embedded experiences] is both given a platform from which to be voiced, and equally important, amplified in ways that are heard” [3]. We hope and expect that this book will continue its journey among change agents, where it will continue the dialogical process between silence, voice and vocabulary. 

## 6. Discussion

### 6.1. Summary of Empirical Findings

In our photovoice project, that was part of a broader PHR process, we collectively unraveled the complex interplay between gendered (theme 1), classed (theme 2) and racialized (theme 3) inequities in relation to the health and wellbeing of paid care workers in long-term care. Living up to photovoice’s and PHR’s emancipatory intentions, we particularly aimed to foster critical consciousness about these structural embedded experiences in dialogue with community researchers. 

Collectively reflecting upon our process, we came to understand that critical consciousness emerged from an iterative and dynamic process between silence, voice and vocabulary. This has implications for PHR scholars in general and for photovoice facilitators in particular. Our reflections illustrate that facilitators need to be sensitive to the different meanings silence can have in photovoice projects. Silence can be the starting point for finding voice, but it can also signal unsafety as the result of silencing acts. Engaging with critical theories, such as intersectionality, played a pivotal role in developing critical consciousness. Theories provide a vocabulary that enabled participants in understanding their experiences as structural and allowed them to voice their critical perspectives to each other, to the researchers and to change agents. Intersectionality, in particular, enabled the unraveling of the complexity and intersecting nature of these inequalities, and this understanding sparked collective action. We learned that engaging with theories requires relational and ethical sensitivity from photovoice facilitators in a horizontal and dialogical manner, which includes being responsive to the vocabularies of community researchers/participants and more. We will discuss our findings in relation to literature on silence, voice and vocabulary below. 

### 6.2. Vocabularies Are Essential for Epistemic Justice

In our reflections, we illuminate the importance of theories as they support the critical consciousness of community researchers in particular. The importance of these vocabularies resonates with the work of philosopher Miranda Fricker on epistemic injustice [24]. PHR scholars, including photovoice scholars, are increasingly recognizing epistemic justice as an important aspect of social justice [61]. Fricker [24] argues that epistemic injustice can occur in two different ways. 

First, testimonial (or: witnessing) injustice occurs when someone is not acknowledged as a reliable knower; their knowledge is ignored, made irrelevant, or judged as untrustworthy because of who they are. This happens, for example, when clients’ experiential knowledge is seen as less credible than professional knowledge [61]. Second, hermeneutic injustice occurs when marginalized groups do not have access to a vocabulary that enables them to interpret their experiences and/or to describe them these as unjust, oppressive or illegal practice. Hermeneutic injustice happens when certain groups do not have equal access to institutions that provide such vocabularies, including academia, political parties, governmental institutions, literature or mainstream media. As a consequence, group members are “*more likely than others in a position (…) where they do not have the concepts or interpretative tropes to render their experiences intelligible to others, possibly even to themselves*” [24] (p. 257). In our photovoice project, we realized that making photographs and organizing a dialogue is not enough to foster hermeneutic justice. We need access to interpretative tropes, as Fricker has argued. 

### 6.3. Engaging with “Interpretative Tropes” Requires Relational Sensitivity

Therefore, in this project, we learned that as facilitators we also needed to take up space for our own expertise and knowledge in dialogue with community researchers, granting them access to these interpretative tropes. In our project, the academic researchers actively provided knowledge on gender, class, race and intersectionality theory. At the same time, facilitators need to stay attuned to interpretative tropes that are introduced by the community researchers themselves or by societal discourses. In our project, such narratives were provided by the yellow-vest and BLM movements. We conclude that photovoice facilitators should not shy away from engaging with these vocabularies, as these are necessary for hermeneutic justice. However, at the same time, engaging with theories requires reflexivity and relational sensitivity from facilitators to not compromise participants’ witnessing justice. Fricker [24] (p. 84) describes this virtue as “testimonial sensibility” which “*enables the hearer to the word of another with the sort of critical openness that is required for a thoroughly effortless sharing of knowledge”.* In other words, developing critical consciousness is a reciprocal learning process. 

### 6.4. Intersectionality Provides Essential “Interpretative Tropes” for Hermeneutic Justice

Another lesson learned from this project is that intersectionality provided essential interpretative tropes to foster hermeneutic justice. The empirical findings from our photovoice process captured in the portraits and book “*What You Don’t See*” show how the health of care workers was shaped at the intersection of gender, class and race. Intersectionality enabled us to unravel these multiple and interlocking inequities within our photovoice process, and it gave us a vocabulary to talk about these inequities. Moreover, we observed that some single-axe analyses such as a “gender perspective” did not resonate with participants, as has been described by other intersectionality scholars [36]. Seeing oppression at an intersection of gender and class sparked activism and collective action among the participants, as it became visible in the op-ed. Over the course of our project, we unraveled how care workers’ health was shaped at the intersections of gender, class and race. Intersectionality helped to reveal these intersections and fostered critical consciousness about complex social inequalities. 

### 6.5. Learning to Listen to Silence

In addition, we paid attention to the many meanings of silence in our research project. Our reflections resonate with the literature on silence in the research process. Scholars have critiqued simplistic (or “thin”) conceptions of voice that are “*focused on explicit utterances and their intelligibility to others*” and plea for a “thick” conception of voice that includes “*a thick interpretative discernment of utterances, silences and bodily expressions*” [62] (p. 2). Carnivale and others redirect our attention to the importance of silences in relation to voice [63,64,65]. Silences, gaps and omissions can lead the way to untold stories, conceptualized as “shadow stories”, which remain hidden behind the spoken narrative [66,67]. Others challenge the assumed equation between voice and power and between silence and oppression [50]. Such an equation is theoretically problematic and empirically untrue and obscures the many possibly empowering meanings that silence can hold [50,63,64,65,66,67]. In addition, Malhotra particularly problematizes the equation between voice and power because “*the burden of social change is placed upon those least empowered to intervene in the conditions of their oppression”* and state that such an equation shifts *“the focus away from the labor that might be demanded of those in positions of power to learn to listen to subaltern inscriptions—those modes of expression that are often interpreted as “silence”*” [50] (pp. 1–2). 

In relation to our own research project, we recognize how silence could be a starting point for finding voice and could, thus, be empowering. However, the community researchers also experienced how they were silenced by the researchers and by change agents. Although they felt empowered by developing critical consciousness in our project, they often felt silenced at their workplace and in dialogue with change agents. Empowerment within the research team, which became a community in itself, does not necessarily translate to other life domains. In line with Malhotra and heeding the call of our community researchers, our project should also be understood as an invitation for change agents to listen to the things that are not (easily) said when we speak about the health and wellbeing of care workers, such as their experiences of poverty and racism. 

### 6.6. Speaking about Oppression Is Painful and Not “Positive” but Is Essential for Hermeneutic Justice

In our photovoice project, rendering social experiences understandable to ourselves was not the biggest bottleneck. However, we did experience how speaking about (multiple and interlocking) systems of oppression was in particular a painful “interpretative trope” to others. This became tangible in tensions within our own research team (in relation to gender and race) as well as in relation to the steering group (in relation to gender/class/race). Interpretative tropes such as gender, class and race are often contested and can be hard to acknowledge in particular by those in a privileged position. This has since long been described by critical race scholars, such as political philosopher Charles Mills [68]. His concept of ‘epistemic ignorance’ describes white people’s epistemological inability to see and acknowledge racism. In addition, with the inability to see inequality, we also observed that norms of “positivity” can turn into silencing discourses when raising your voice about injustice is not listened to because it is not “constructive” [69,70]. We recognized this dynamic in the reactions to our op-ed in specific in response to the project findings as presented in the photographs of the community-researchers at the end of the project, as well as in the community researchers’ experiences in the workplace in general. 

### 6.7. Strengths and Limitations of Our Study

In this article, we reflected upon a photovoice project as a team of academic researchers, community researchers and a professional photographer. In doing so, we have been able to capture a wide array of voices and perspectives. Fostering reflection together with all participants contributes to the quality of the study and to ethical research practice. However, we also described how the white, male community researcher withdrew from the study, as he felt his perspective was not sufficiently addressed. As a consequence, the reflections in this paper mainly focused on women’s experiences, while we know that men’s experiences are differently shaped by norms around masculinity [43]. Photovoice is highly local and contextual. However, dynamics within our photovoice project resonate with empirical findings in our broader PHR project “Negotiating Health” [42,43,44,45], which suggest that findings can be transferable in the Dutch context.

## 7. Conclusions

In this paper, we have described how critical consciousness in photovoice, as part of a broader PHR project, springs from an iterative process between silence, voice and vocabulary. This has several implications for participatory health scholars in general and for photovoice scholars in particular. First, we conclude that facilitators need the courage and sensitivity to engage with “vocabularies” from academic theories or emancipatory movements. These vocabularies support critical consciousness and are essential for epistemic justice in photovoice. Second, we conclude that intersectionality is a useful interpretative framework within PHR and photovoice. Intersectionality enables critical consciousness about complex and intersecting inequities and supports collective action. Third, we conclude that PHR and photovoice should go beyond fostering voice to those whose experiences are marginalized. The responsibility for social change lies with those in power who can learn to listen to care workers’ silences, particularly on issues that are not easily said in relation to their health and wellbeing, such as gendered poverty and racism. 

## Figures and Tables

**Figure 1 ijerph-19-05570-f001:**
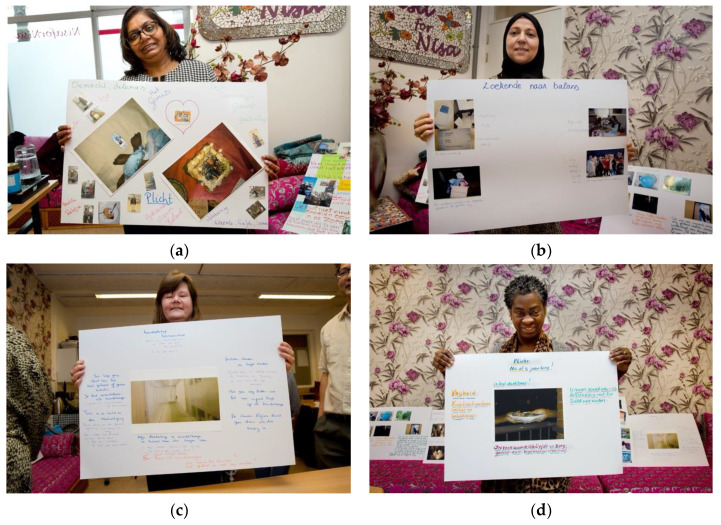
(**a**–**d**) Portrait of community researchers with chosen photographs.

**Figure 2 ijerph-19-05570-f002:**
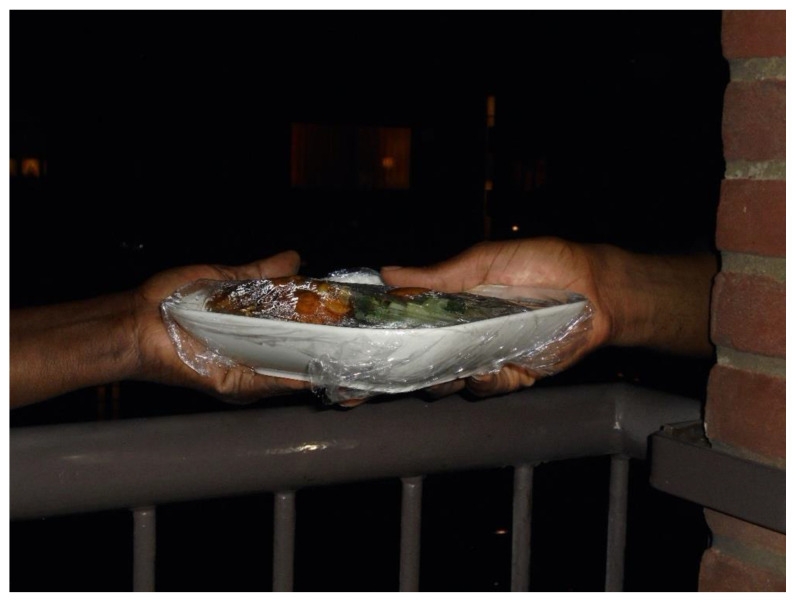
Photograph of community researcher handing over food to her son, titled: “my duty, unconditional love and care, thankful”.

**Figure 3 ijerph-19-05570-f003:**
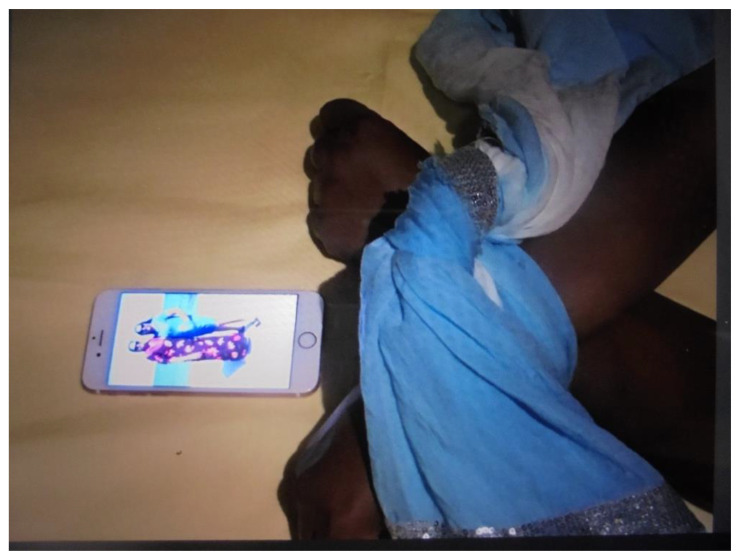
Photograph of a community researcher whose hands are tied while looking at a picture from her mother who lives in Surinam, titled: “missing out/the loss”.

**Figure 4 ijerph-19-05570-f004:**
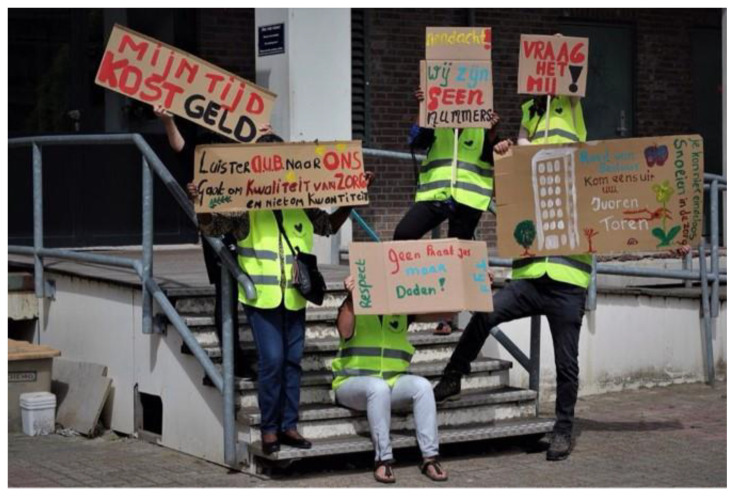
Picture of community researchers with protest signs and yellow vests.

**Figure 5 ijerph-19-05570-f005:**
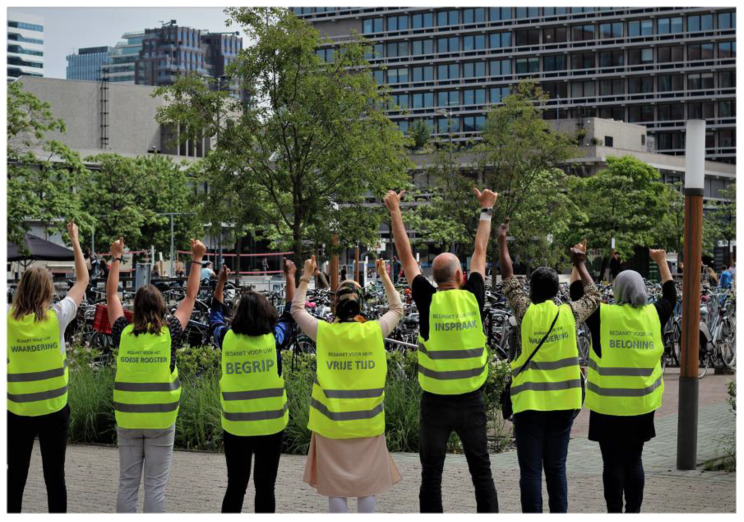
Picture of community researchers with yellow vests.

**Figure 6 ijerph-19-05570-f006:**
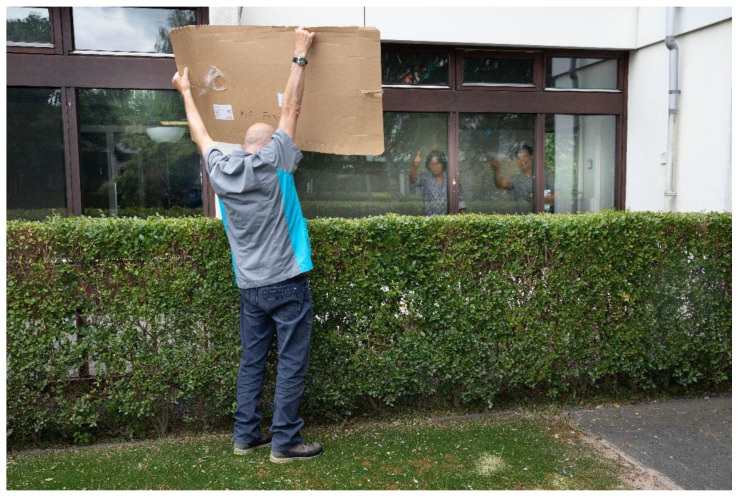
Pictures of community researcher doing paid care work with protest sign.

**Figure 7 ijerph-19-05570-f007:**
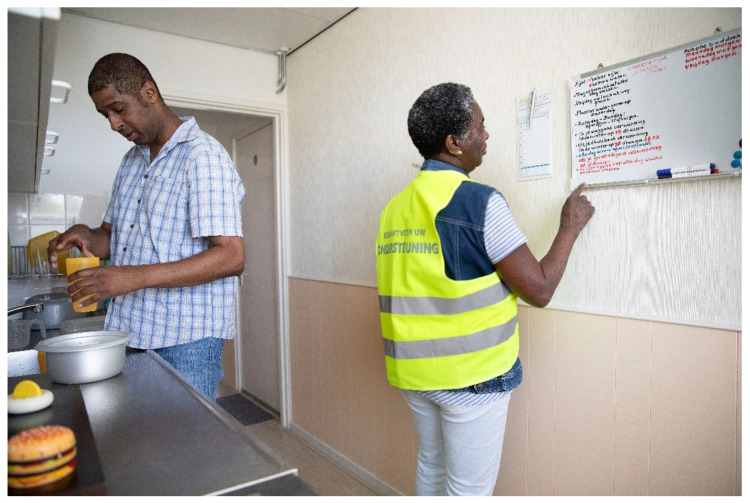
Picture of community researcher doing unpaid care work wearing yellow vest.

**Figure 8 ijerph-19-05570-f008:**
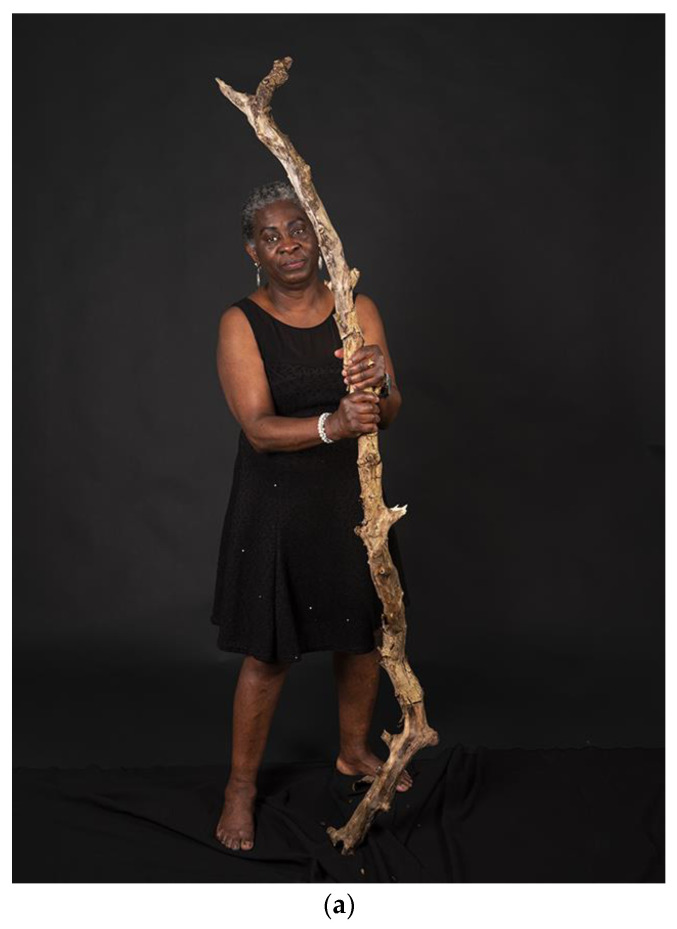
(**a**–**d**) Portraits of community researchers in book “*What You Don’t See”*.

**Figure 9 ijerph-19-05570-f009:**
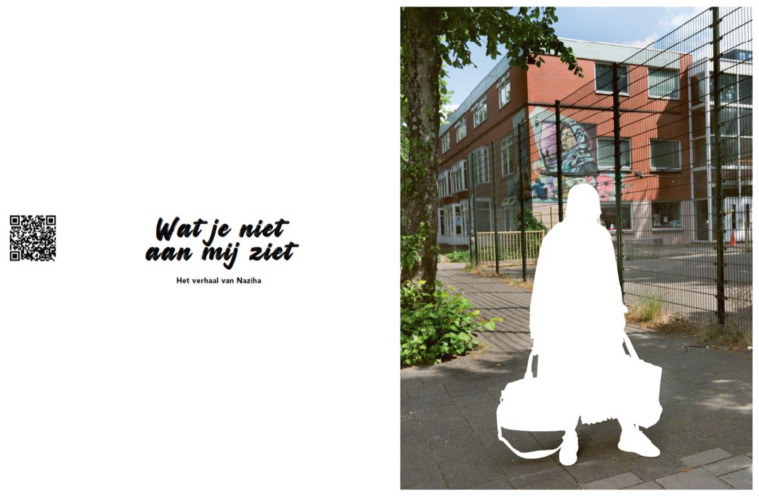
Images in book *“What You Don’t See”*.

**Table 1 ijerph-19-05570-t001:** Phases of PHR project Negotiating Health (2018–2022).

Phase	Year	Activities	Participants	Results	Typology of Hermeneutic Understanding	Critical Lens
1	2018–2019	Photovoice (*n* = 10 meetings)	10 (un)paid caregivers	Article in journal for professionals in the health and social care domain [40]	Academic researchers reflecting about participants’ photographs and narratives	Gender
2	2019–2021	Photovoice	5 co-researchers	Op-ed in national newspaper [41]	Dialogue between co-researchers, photographer and academic researchers	Gender/Class
3	2019–2021	PHR projects	5 co-researchers	Scientific article #1 [42] Scientific article #2 [43] Scientific article #3 [44] Scientific article #4 [45]	Academic researchers and co-researchers reflecting about respondents in the qualitative sub-studies of Negotiating Health	Gender/Class/ Race/Disability/Sexuality
4	2019–2021	Photovoice	5 co-researchers	Portraits and Book	Co-creation of portraits and book	Gender/Class/ Race/Disability/Sexuality
5	2021–2022	Dialogue and Action	4 co-researchers	Dialogue meetings with change agents Book presentation	Dialogue with change agents	Gender/Class/Race/Disability Sexuality

## Data Availability

The data presented in this study are available upon request from the corresponding author. The data are not publicly available for privacy and ethical reasons.

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
