# Peer review of "Navigating Voice, Vocabulary and Silence: Developing Critical Consciousness in a Photovoice Project with (Un)Paid Care Workers in Long-Term Care"

_ijerph, 2022, doi:10.3390/ijerph19095570_

Round 1

Reviewer 1 Report

Thank you for the opportunity to review this paper. I enjoyed reading it.

This is a well-written, well-explained and original piece of study. The authors argue how a 4-year long photovoice project helps to develop critical perspectives on structural systems of oppression of the participants  - paid and unpaid care-givers in residential long-term care. They critically reflected upon how their experiences are shaped by structural inequalities, including gender, class, ethnicity and age. The authors explain well the complexity of an iterative process between silence, voice and vocabulary and how their findings relate to critical theories, such as intersectionality.

Perhaps the authors would like to elaborate a bit more (unless this is out of scope in this paper and was not taken into consideration in this study) on the age of the paid and unpaid care-givers who took part in the project, and how age dynamics contribute to silences, gaps and voices.

The authors state that ‘silence could be a starting point for findings voice, and could thus be empowering’ (p. 24) – the importance of silences and gaps is also explored in narrative gerontology, which is related to the authors' research. Having a look at these (and similar) papers, which focus on the untold stories and careful listening that include omissions that form a type of ‘shadow story,’ could enrich the authors' research:

-“Shadow stories” in oral interviews: Narrative care through careful listening. By Kate de Medeiros and Robert L. Rubinstein. http://dx.doi.org/10.1016/j.jaging.2015.02.009

-The Importance of Untold and Unheard Stories in Narrative Gerontology: Reflections on a Field Still in the Making from a Narrative Gerontologist in the Making. By Bodil Hansen Blix. Narrative Works: Issues, Investigations, & Interventions 6(2), 28–49

The authors mention briefly that there are some differences in the responses and ‘silences’ of male and female paid and unpaid care-givers. I wonder if they have encountered any more significant differences that are worth acknowledging (related to gender and age).

In the future, it would be interesting to carry out a similar project with the residents of long-term care homes and examine the power of their unvoiced narratives through a photovoice project in relation to their wellbeing and quality of life, as well as gender, race and age dynamics.

Good luck with your future research!

Reviewer 2 Report

The  paper deals with the topic of “Navigating voice, vocabulary and silence: developing critical consciousness in a photovoice project about the health of (un)paid care workers in residential long-term care”.

In general, this is an interesting paper which presents a very broad spectrum of analysis in regards with this issue. There are few aspects which the authors should take into account and revise:

  1. The authors should extend the main insights and conclusions which appears in the abstract.
  2. The authors use the “intersectionality” term – but do not emphasis it well. There is a need to elaborate its’ direct meaning and implication based on the theory and the results.
  3. What are the main objectives of the paper? There is a need to present and explain the objectives in the Introduction
  4. Section 2 – there is a major need to extend the presentation of the theoretical framework, through an in-depth and detailed proposal of scientific literature that examines all the theoretical issues that are under discussion in the current scientific analysis.
  5. The authors need to present and discuss possible research limitations and discuss the research implications that result from their potential existence.

Reviewer 3 Report

he article portrays an innovative topic that is very relevant.

The context of long-term care is related with major public health concerns and of great interest to Public Health.

However the ethical approval for the study and for the pictures are mandatory.

The Title is too long.

Picture in a different language

Pictures should me selected, too many.

Participatory health research is key aspect of this manuscript should be more explore in discussion

Conclusions should be more related with the results and without references

The impact of the article should be mentioned with recommendations.

The Equator Checklist should be checked.

Limitations should be mentioned.

Author Response

This manuscript is a resubmission of an earlier submission. The following is a list of the peer review reports and author responses from that submission.